# Applying historical data in a nonlinear mixed-effects model can reduce the number of control rats required for calculation of the relative potency of insulin analogues

**Emilie Prang Nielsen**[1☯]*, **Søren Andersen**[1☯], **Christian Lehn Brand**[2☯], **Susanne Ditlevsen**[3☯]

**1** Biostatistics, Novo Nordisk A/S, Søborg, Denmark, **2** Diabetes Pharmacology, Novo Nordisk A/S, Måløv, Denmark, **3** Data Science Laboratory, University of Copenhagen, Copenhagen, Denmark

☯ These authors contributed equally to this work.
* emilie.nielsen@hotmail.com

**Data Availability Statement:** The dataset is available as supporting information S1 Dataset.

## Abstract

This paper examines how to reduce the number of control animals in preclinical hyperinsulemic glucose clamp studies if we make use of information on historical studies. A dataset consisting of 59 studies in rats to investigate new insulin analogues for diabetics, collected in the years 2000 to 2015, is analysed. A simulation experiment is performed based on a carefully built nonlinear mixed-effects model including historical information, comparing results (for the relative log-potency) with the standard approach ignoring previous studies. We find that by including historical information in the form of the mixed-effects model proposed, we can to remove between 23% and 51% of the control rats in the two studies looked closely upon to get the same level of precision on the relative log-potency as in the standard analysis. How to incorporate the historical information in the form of the mixed-effects model is discussed, where both a mixed-effect meta-analysis approach as well as a Bayesian approach are suggested. The conclusions are similar for the two approaches, and therefore, we conclude that the inclusion of historical information is beneficial in regard to using fewer control rats.

## 1 Introduction

Every year, thousands of animals are used in preclinical studies, for example when developing a new drug. These studies are so-called in vivo or within the living, which means that a whole living animal is used to collect data in support of the safety of a new treatment. When using large numbers of animals such as rats and mice, both ethical and economical issues arise. Some of these issues might be avoided by using information gained from historical studies, which could result in reducing the number of laboratory animals. The aim of this paper is to investigate how many control animals can be spared if we make use of information from historical studies in a series of hyperinsulinemic glucose clamp studies in rats. The reduction of

**Funding:** Susanne Ditlevsen was supported by Novo Nordisk Foundation NNF20OC0062958 and Independent Research Fund Denmark | Natural Sciences 9040-00215B.

**Competing interests:** Emilie Prang Nielsen, Søren Andersen and Christian Lehn Brand are employed by Novo Nordisk and hold shares in the company. This does not alter our adherence to PLOS ONE policies on sharing data and materials.

control rats is based on obtaining the same standard error on the relative log-potency as when disregarding historical data. Three different statistical modelling approaches are compared, in which historical data is either ignored or taken into account. The historical data is incorporated both by a mixed-effects meta-analysis approach and a Bayesian approach. Results on the standard error for the relative log-potency using only current data is compared to the results from the approaches incorporating historical information.

The methods are applied to a dataset provided by Novo Nordisk A/S, consisting of 59 preclinical hyperinsulinemic glucose clamp studies in rats conducted from 2000 to 2015. The dataset is available in S1 Dataset. The purpose of the studies are to investigate new insulin analogues for diabetics, typically by comparing the analogues to a control insulin, human insulin (HI). HI is repeated in most studies, giving in total 885 control rats in 52 studies, and all the studies during the 15 years are performed based on the same protocol (with one amendment) and same rat species. This forms the basis for investigating if some of these control rats could have been spared by including data from previous studies (both historical control rats and insulin analogues), using results on the standard error for the relative log-potency based on the mixed-effects meta-analysis approach and the Bayesian approach as compared to the standard approach ignoring historical information. This is done through a simulation experiment, where the effect of a reduction in the number of control rats is examined.

The use of historical data is not a new phenomenon, and it has been investigated in the literature. In [1] they construct a mixed-effects model including historical controls and conclude that this approach can be useful in regard to increasing precision, given a degree of similarity between the studies. A large similarity between historical and current control data is clearly an important factor for incorporating historical data successfully, which we also investigate carefully in the 15 year series of hyperinsulinemic glucose clamp studies used for this article. Mostly in the literature, only historical controls are incorporated in the model (an approach used in both [1, 2]), but due to our large amount of historical insulin analogue data and the structure of our model, we can benefit from constructing a mixed-effects model including both the historical control rats (most importantly) but also the historical insulin analogues in order to increase precision even further. In [3], see also references therein, they incorporate the historical data using a Bayesian approach, and in [4] several Bayesian methods to include historical information are compared, where they conclude that it is important to estimate and allow for the heterogeneity between trials in the analysis. A Bayesian approach is also investigated in this article, and an important distinction is that we compare the results from the Bayesian approach and the mixed-effects meta-analysis approach through a simulation experiment, which is not commonly seen in the literature. However, see [4] for a simulation study to compare Bayesian methods.

The outline of the paper is as follows; a short description of the preclinical hyperinsulinemic glucose clamp studies and the data used is presented in Section 2. Section 3 introduces the different modelling approaches as well as the model specifications. In Section 4, the results from a simulation experiment are presented, and it is examined closely whether or not the approaches proposed might result in using fewer control rats. The methods will be discussed briefly in Section 5, and Section 6 wraps up the findings.

## 2 Data

When a new insulin analogue is characterized preclinically with respect to its pharmacological properties, measurement of the in vivo potency relative to a well-known reference insulin is important in order to be able to predict the doses required to control blood glucose levels in people living with diabetes.

We here present data produced over a 15 year period, during which the potencies of 69 different insulin analogues were measured relative to human insulin (HI) [5] using the hyper-insulinemic euglycemic clamp technique in a healthy laboratory rat strain. A total of 59 independent clamp studies were performed and included a total of 2,567 rats of which 885 served as HI controls. In some of the studies, more than one insulin analogue was included, which results in a larger number of insulin analogues compared to number of studies. Different molar infusion rates (i.e. doses in pmol/kg/min) of the insulin molecule in question were administered intravenously in individual rats. Plasma glucose levels were maintained (clamped) at a predefined target value by means of an intravenous infusion of glucose, which was adjusted regularly based on the fluctuations in plasma glucose concentrations measured at regular intervals (typically every 10th min) during the clamp experiment. The structure of the study is shown and explained in Fig 1.

By clamping plasma glucose at euglycemia during the constant insulin infusion even very high doses of insulin can be tested without activating the counter regulatory responses (e.g. glucagon, catecholamines, corticosterone) due to hypoglycaemia, which would otherwise underestimate the effect of insulin. The glucose infusion rate (GIR in mg/kg/min) required to

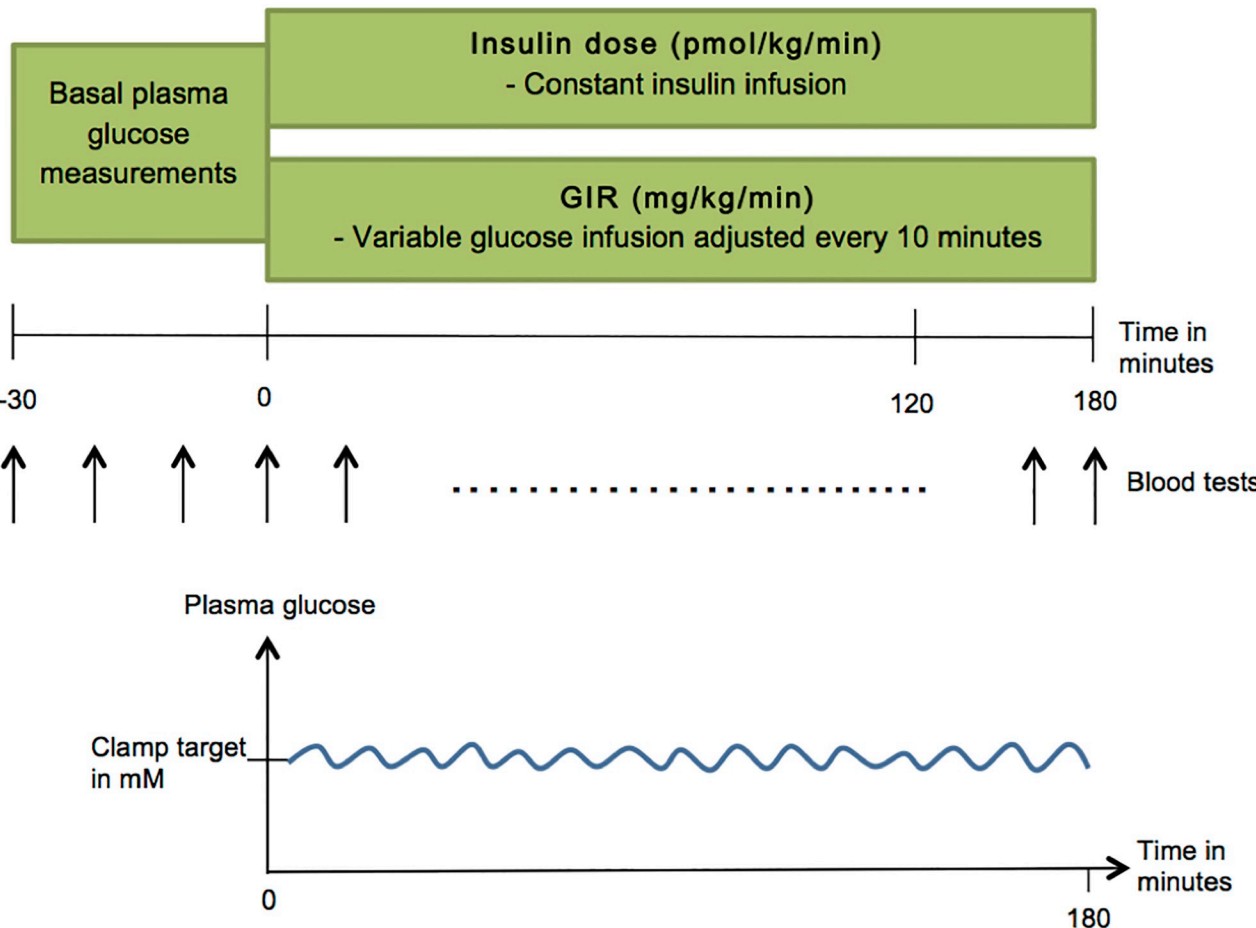

**Fig 1. The experimental clamp study day.** Basal plasma glucose values are measured 30 minutes before time zero, where the study starts. The rat is given a constant infusion of insulin and a variable infusion of glucose, where the latter is adjusted every 10 minutes based on the plasma glucose concentration, which is found by blood tests. The goal is to maintain a constant level (the clamp target) of plasma glucose.

counteract the glucose lowering effect of insulin then becomes a quantitative measure of the insulin effect.

Due to pharmacokinetic properties of the insulin analogues, they were administered as a primed continuous infusion lasting for up to 5 hours in order to approach steady state in both plasma insulin concentrations and GIR. The average GIR during the last 60 minutes of the experiment in each rat was used as the quantitative measure of the effect of insulin. Note that all 59 studies conducted during the 15 years were performed based on the same protocol (with one amendment, see below), same control insulin (HI) and same rat strain. During 2008, it was decided that the previous clamp target had been too high based on the measurements of the rats' basal plasma glucose levels (the only amendment in the protocol). Therefore, from a basal plasma glucose level of 6.2 mM, in 2008 the clamp target was changed to 5.7 mM, which is also taken into account in the mixed-effects model presented in Section 3.2.2. A more detailed description of the clamp method in rats including the preceding surgical insertion of permanent catheters for infusion and blood sampling can be found in [6].

## 2.1 Choices of doses, design and logistics of a typical clamp study

The choice of doses of a new analogue to be tested were based on prior in vivo and in vitro data as well as on the growing experience with the different families of analogues over the years. The less prior data and knowledge about a new analogue the more doses had to be tested in order to ensure that the effect (GIR) was measured sufficiently. The full dose response curve for HI has been established more than once in this framework, thus, in general fewer doses of HI than the analogue were needed.

As the clamp experiment as well as the preceding surgical catherization and a post-surgical restitution period (9–11 days) is laborious, cumbersome and requires 2–3 operators to be next to the animals during the entire clamp experiment, it takes up to several weeks to perform a clamp study. For example, testing three doses of a new analogue against three doses of HI each in six rats took in average three weeks. Using this example, the three doses of each of the two insulin molecules were stratified on each experimental day over the three weeks in order to distribute the day-to-day variation equally over the three weeks. The individual rat receiving a predetermined insulin molecule and dose was randomly selected among the surgically prepared group of rats.

## 3 Methods

The experimental procedures involving animals were approved by the national Danish Animal Experiments Inspectorate and the Ethical Review Council at Novo Nordisk A/S. The surgical catheter procedure was conducted under general isoflurane anaesthesia followed by three days of analgesia obtained by daily carprofen injections. At the end of the experiments animals were killed by an intravenous overdose of pentobarbital.

## 3.1 The logistic curve

To relate the response (GIR) to the given dose of insulin, we use a symmetric, logistic model that approximates two parallel straight lines (one for the insulin analogue and one for HI) in the middle dose region. Due to tradition in this setting and to get a better fit to data, we log-transform the doses. Letting $z$ be the non-transformed doses, we define $x = \log(z)$, and these log-transformed doses will be used throughout this paper when fitting the different models. We refer to the log-transformed doses by the term logdose, and the original non-transformed doses by the term dose.

To obtain an average GIR close to zero for an insulin dose of zero, we use the following representation of the symmetric logistic dose-response curve, defined in [7]:

$$f(x) = \frac{d}{1 + e^{-b(x-\log(ED_{50}))}}. \tag{1}$$

where $f(x)$ denotes the average GIR for a given logdose, $\log(ED_{50}) \in \mathbb{R}$ is the value of $x$ where the average GIR is half of its maximal value, $b \in \mathbb{R}$ is proportional to the slope around this point and $d \in \mathbb{R}_+$ is the maximal value of the GIR. If $b > 0$, the curve will be monotonely increasing.

The relative log-potency, $\log \rho$, of the insulin analogue compared to HI is defined as follows:

$$\log \rho = \log\left(\frac{ED_{50}^{HI}}{ED_{50}^{A}}\right) = \log(ED_{50}^{HI}) - \log(ED_{50}^{A}). \tag{2}$$

The terms $ED_{50}^{HI}$ and $ED_{50}^{A}$ denote the doses needed of HI and the insulin analogue, respectively, to obtain 50% of its maximal effect. The smaller the dose needed for this, the more potent the drug is said to be, which is indicated in the top of Fig 2. The relative log-potency is thereby the log of the ratio between these two amounts, which can be written as a difference between the two logdoses, and estimated using (1). This is shown in Fig 2, from which it also becomes clear that we have made an assumption about parallelism, such that the curves for HI and the insulin analogue have the same parameters $b$ and $d$. By making this assumption, we ensure that the estimates for the relative log-potency are valid and interpretable [8]. Note from (2) and Fig 2

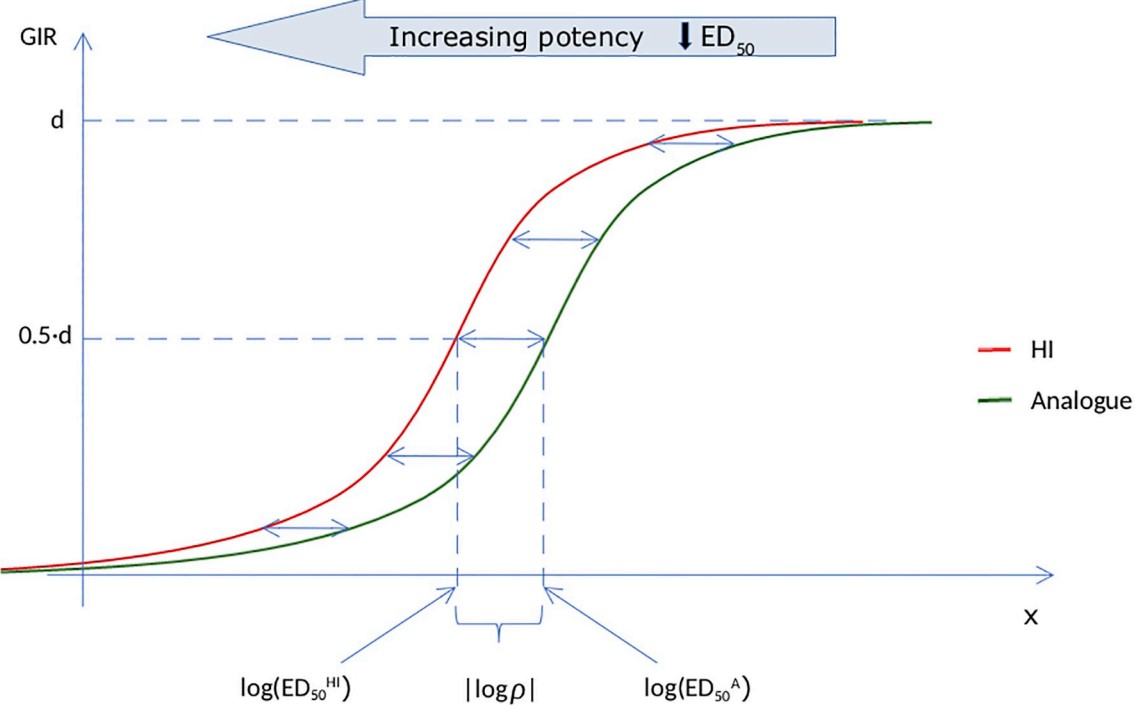

**Fig 2. The logistic model, Eq (1), for the GIR as a function of the logdose of constant insulin infusion for HI and an insulin analogue.** The two curves are parallel, and the relative log-potency, log $\rho$, is defined as the horizontal difference between the curves. The further the curve is to the left, the higher the potency.

that $\log \rho$ is negative if a smaller logdose of HI than the insulin analogue is needed to obtain half of its maximal effect, which would mean that HI is more potent than the insulin analogue.

## 3.2 Modelling approaches

The dose-response relationship is modelled using the logistic curve in Eq (1). The analysis is carried out using three different approaches in which historical data is ignored or taken into account, either by a mixed-effects meta-analysis approach or a Bayesian approach. The parameters are estimated by maximum likelihood or Markov chain Monte Carlo methods, respectively, and the R-code for the model fits can be found in S1 Appendix [9].

**3.2.1 The standard approach: Each study at a time.** Usually, the relative potency of a new insulin analogue compared to HI was investigated by using data only from that particular study. Thereby historical data was ignored, and each study uses its own control group [10]. Using Eqs (1) and (2), the model is as follows: Within each study we assume to have $t \in \{HI, A\}$ dose-response curves ($t$ for treatment), where one insulin analogue is compared with the control insulin, HI. Letting $j = 1, .., n_t$, where $n_t$ is the number of rats for drug $t$, we get the following model [7]:

$$y_{tj} = f(x_{tj}, \beta) + \varepsilon_{tj}, \tag{3}$$

where $x_{tj}$ denotes the $j$th logdose value in the $t$th dose-response curve, $y_{tj}$ is the resulting GIR value for that logdose value, and $\varepsilon_{tj}$ is the measurement error for the response $y_{tj}$. The errors $\varepsilon_{tj}$ are assumed mutually independent and normally distributed $N(0, \sigma^2)$. The unknown parameter vector is denoted by $\beta_t$, and we thus have the following parameters for HI and the insulin analogues respectively:

$$\begin{aligned}
\beta &= (b, d, \log(ED_{50}^{HI}), \log(ED_{50}^{A})) \\
&= (b, d, \log(ED_{50}^{HI}), \log(ED_{50}^{HI}) - \log\rho).
\end{aligned} \tag{4}$$

Here we only use data from one study at a time. Using this approach on each of the historical studies, we got a mean estimate for $\log(ED_{50}^{HI})$ of 3.1 with a standard deviation of 0.76. Due to the variation in the estimates of the potency for HI, including control rats in future studies are needed. However, including also the historical information reduces the amount of control rats needed to obtain the same standard error. In the following, it is examined to what degree information from historical control rats can be used in future studies based on the between study variation.

**3.2.2 A mixed-effects meta-analysis approach.** From Section 2 it is clear that we have a large amount of historical data, which we could potentially include, especially for the control rats on HI. It is inefficient and a waste of data to ignore historical information on controls when comparing with new insulin analogues [10]. In our case not only historical data on control rats, but also historical data on the insulin analogues could be of potential interest, as the model parameters b and d coincide for the two drugs due to the assumption of parallelism (see Section 3.1). However, since the insulin analogue under investigation is generally studied for the first time, it is clearly the inclusion of historical information from the control rats that will be of the highest importance.

The historical data can be taken into account by a meta-analysis approach. The idea behind a meta-analysis is essentially to combine the results of multiple scientific studies, and historical studies provide a solid foundation for a meta-analysis [3]. We therefore aggregate all historical information with data from the study under investigation. Thereby, the number of data points for HI included in the analysis increase, which intuitively leads to more precise results. This method of complete pooling is described in [10].

The drawback of the approach including only fixed effects is that we do not allow for any variation between the studies. Even if the studies are performed using the same protocol and the same types of rats, the batches of rats are ordered for each individual study, and there can be small variations between the batches. Furthermore, the laboratory technicians do not change within a study, but are likely to change from study to study, which may induce additional variation. The model where we include these random variations between studies is a mixed-effects model, and this approach will be referred to as the mixed-effects meta-analysis approach.

By investigating thoroughly which fixed and random effects to include in the model (different model validations, comparing AIC values), we ended up with the following model specification (see Appendix A.3 in S1 Appendix for the model specification in R):

Let $i = 1, .., 59$ denote the index of the study, $j = 1, .., n_i$ denote the observation index for the $i$th study and $a = 1, .., 69$ denote the 69 different insulin analogues. The model for the $i$th study is:

$$
\begin{aligned}
y_{ij} &= f(x_{ij}; \beta, \psi) + \varepsilon_{ij} \\
&= \frac{d + d^{high} \cdot \mathbb{1}_{\{high\}} + A_i}{1 + \exp(-(b + b^{high} \cdot \mathbb{1}_{\{high\}})(x_{ij} - (\log(ED_{50}^{HI}) + B_i - \mathbb{1}_{\{Analogue\}} \cdot \log \rho_a)))} + \varepsilon_{ij},
\end{aligned}
\tag{5}
$$

where $x_{ij}$ denotes the $j$th log-transformed dose value in the $i$th study, $y_{ij}$ is the resulting GIR value for that logdose, and $\varepsilon_{ij}$ is the measurement error for the response $y_{ij}$. $A_i$ and $B_i$ denote the random effects for $d$ and $\log(ED_{50}^{HI})$ respectively. The errors $\varepsilon_{ij}$ are assumed mutually independent, normally distributed $\mathcal{N}(0, \sigma^2)$ and independent of the random effects $(A_i, B_i)$. Furthermore, $\mathbb{1}_{\{Analogue\}}$ is an indicator function that takes the value one if insulin analogue and the value zero if HI. Finally, $\mathbb{1}_{\{high\}}$ takes the value one for the studies with the high clamp target, and zero for the low target. With this model specification, we follow Eqs (1) and (2) from Section 3.1.

The unknown fixed parameter vector, $\beta$, is given by:

$$
\begin{aligned}
\beta &= (b, b^{high}, d, d^{high}, \log(ED_{50}^{HI}), \log \rho_a) \\
&= (b, b^{high}, d, d^{high}, \log(ED_{50}^{HI}), \log \rho_1, ..., \log \rho_{69}),
\end{aligned}
\tag{6}
$$

where $d \in \mathbb{R}$ and $b \in \mathbb{R}_+$ are the parameters for a clamp target 5.7 mM. The parameters $b^{high} \in \mathbb{R}$ and $d^{high} \in \mathbb{R}$ denote the difference in these parameters from a target of 5.7 mM to the higher target of 6.2 mM. The parameter $\log(ED_{50}^{HI}) \in \mathbb{R}$ is the logdose when the average GIR is half of its maximal value and is the same for the two targets. Furthermore, $\log \rho_1, .., \log \rho_{69}$ denote the relative log-potencies compared to HI for the 69 insulin analogues. Lastly, as the random effects belonging to each study are assumed independent, their distribution is specified as follows:

$$
(A_i, B_i) \sim \mathcal{N}(0, \psi),
\tag{7}
$$

where the covariance matrix in the normal distribution is given by

$$
\psi = \begin{pmatrix} \psi_d^2 & 0 \\ 0 & \psi_{\log(ED_{50}^{HI})}^2 \end{pmatrix}.
\tag{8}
$$

Here, the idea is that the historical control groups are exchangeable with the current control group from the study under investigation [10]. This model assumes a common behaviour of HI across the studies, inducing a random between study variation by including random effects

on some of the model parameters. One could argue that within study variances, $\sigma^2$, could be different. However, we thoroughly studied the assumption about the same $\sigma^2$ across all studies (and within studies between dose levels). Due to the same hyperinsulinemic glucose clamp setup over time, the within study variances for the different studies were very similar. Note that this assumption also plays an important role when performing the Bayesian analysis in Section 3.2.3.

The estimates, standard errors and 95% wald confidence intervals for the parameters from the model in Eq (5) are reported in Table 1. For simplicity, we have chosen only to report the relative log-potency for two randomly chosen studies, namely study 040801 (insulin analogue 7) and study 060906 (insulin analogue 32), which are also the studies we focus on in Section 4. The parameter estimates are used as the foundation for both the Bayesian approach (Section 3.2.3) and the simulation experiment described in Section 4.1. The results for all the 69 insulin analogues are pictured in Fig 3.

The model fits stratified by study and HI or an insulin analogue are shown. In some studies, only insulin analogues were investigated, and in a few studies, only HI appears. However, in most studies, both drugs were investigated together. Looking at the fits across the studies, there seems to be no extreme "outlier studies", and it is actually difficult to even tell the difference between the population (only fixed effects) and the within study (including random effects) predictions in many cases. Even in the studies with only a few doses, for example study 010303 and 130602, this model still seems reasonable. Modelling it this way, we are even able to get an estimate for the relative potency in the studies where HI is not used as control.

**3.2.3 A Bayesian analysis.**   When considering an unknown future study, using the mixed-effects meta-analysis approach, we would have to aggregate data from this study with the historical studies and repeat the analysis again. This can of course be done, and there is little reason to suspect that this will be computationally heavy. However, it also brings up the question if we are able to include the historical information in an alternative way which might be neater and more transparent. For this purpose we use a Bayesian approach.

The overall idea follows Bayes' theorem, namely that we have a prior belief about the distribution of the parameters, which we then revise in the light of the evidence (the likelihood of the data) in order to produce a posterior distribution for the parameters [10]. Following this idea, we need a prior belief about the parameters, and if the results from similar historical studies are available, these can be used as the prior distribution [10]. Consequently, the more

**Table 1. Estimates, standard errors and 95% confidence intervals for the parameters from the mixed-effects model.** Only the relative log-potency for two insulin analogues are presented here, $\log \hat{\rho}_7$ (study 040801) and $\log \hat{\rho}_{32}$ (study 060906).

| | Estimate | Std error | 95% CI |
|---|---|---|---|
| $\hat{b}$ | 1.750 | 0.042 | (1.667;1.832) |
| $\hat{b}^{high}$ | -0.235 | 0.059 | (−0.348;−0.121) |
| $\hat{d}$ | 35.113 | 0.477 | (34.191;36.036) |
| $\hat{d}^{high}$ | 2.985 | 0.787 | (1.463;4.506) |
| $\log(\hat{ED}_{50}^{HI})$ | 2.804 | 0.028 | (2.750;2.858) |
| $\log \hat{\rho}_7$ | -0.618 | 0.090 | (−0.794;−0.442) |
| $\log \hat{\rho}_{32}$ | -0.414 | 0.122 | (−0.649;−0.179) |
| $\hat{\psi}_d$ | 1.414 | - | (0.877;2.279) |
| $\hat{\psi}_{\log(ED_{50}^{HI})}$ | 0.122 | - | (0.089;0.168) |
| $\hat{\sigma}$ | 3.726 | - | (3.622;3.833) |

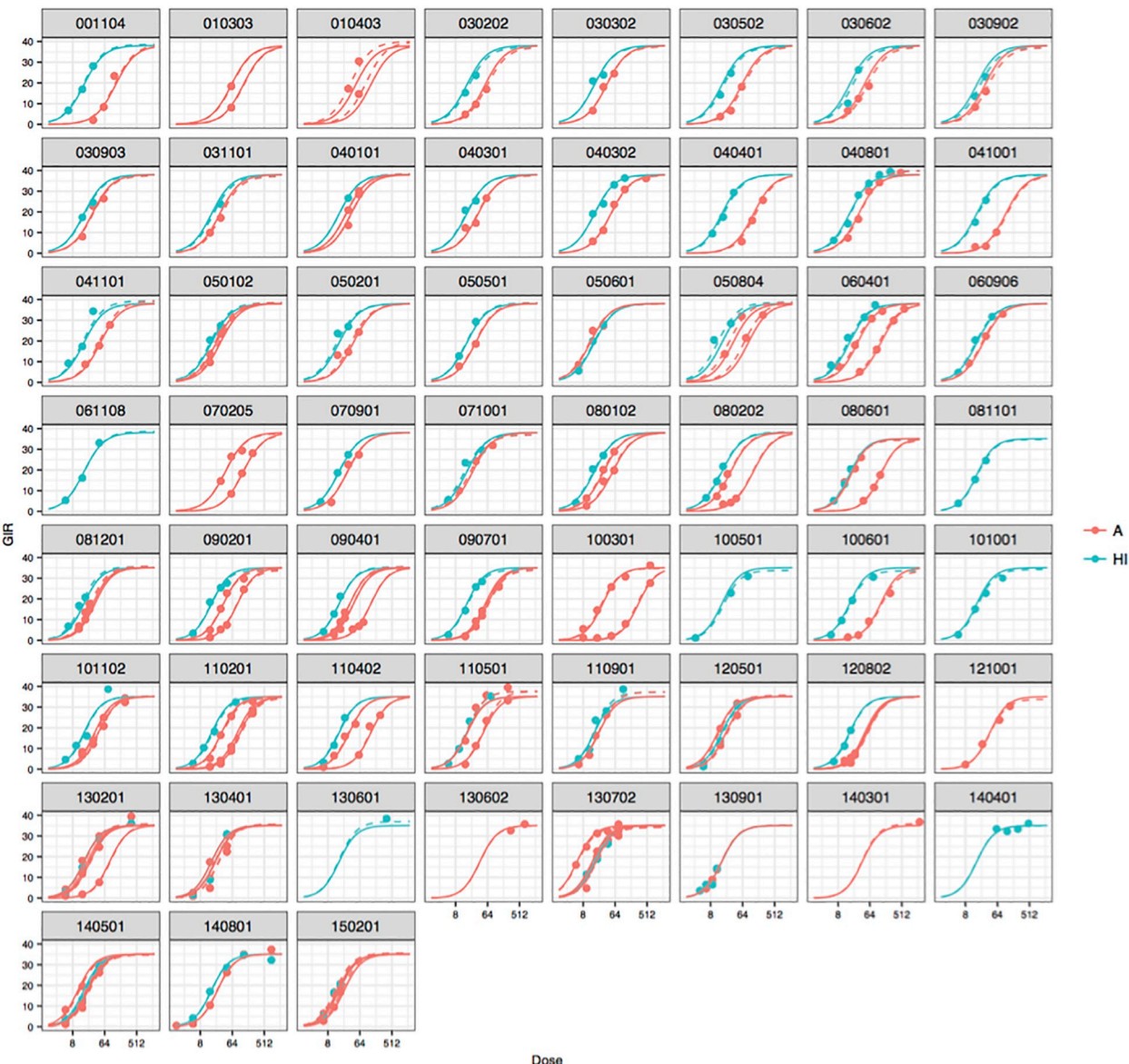

**Fig 3. The fits of the mixed-effects model by study and HI or insulin analogue.** The solid lines are the population predictions, and the dotted lines are the within-study predictions. The points are the observed GIR values.

similar the historical studies behave, the better this prior will be, as we are then quite sure that the new study under investigation will adopt a similar behaviour as the historical studies. Other approaches for choosing a proper prior are described in the literature, but will not be elaborated on here.

Like in the standard approach, we consider only data from the study under investigation, but the difference is that now we will incorporate the historical information in the form of a prior belief for the parameters. This prior consists of precisely the estimates from the mixed-effects model presented in Table 1 and the corresponding distribution presented in Section 3.2.2. As underlined in [10], it is important that we also include the random between study

variation on the parameters in the prior, otherwise we would obtain too little variation on these parameters when considering a new study.

We choose an uninformative prior for the relative log-potency, namely $\log \rho \sim \text{unif}(-3, 3)$. By letting this prior be uniform in a quite large interval, we thereby assume that we do not have any information on the insulin analogue under investigation beforehand [10]. Following [3], we choose a vague inverse gamma prior for the within study variance, $\sigma^2$, such that the precision $\frac{1}{\sigma^2} \sim \Gamma(0.001, 0.001)$. This implies that $\log(\sigma^2)$ will be uniformly distributed, and hence this corresponds to a non-informative prior [10]. Another possibility was to use the estimated $\sigma^2$ from the mixed-effects model presented in Table 1 in the prior, and this gave very similar results. Concerning the remaining parameters, we use a multivariate normal prior with the estimates from the mixed-effects meta-analysis from Section 3.2.2 as mean vector, and the variance-covariance matrix is found by using the estimated correlation matrix for these parameters. To explore the robustness of this method, we tried to increase the variance on the prior distribution, which did not change the results noticeably. The details for the Markov chain Monte Carlo (McMC) sampler used for this analysis (number of chains, iterations and so on) can be found in Appendix A.4 in S1 Appendix, and convergence of the MCMC sampler was obtained.

In Section 4.3, the results from the corresponding posterior distribution of the parameters are compared with the results obtained by performing the mixed-effects meta-analysis.

## 4 Reducing the number of control rats

We have now presented three different modelling approaches and these are applied focusing on two randomly chosen historical studies, study 040801 and 060906. The results are compared for the parameter of interest, the relative log-potency $\log \rho$. This is done through a simulation experiment.

### 4.1 A simulation experiment

The first step in this simulation experiment is, for each of the chosen historical studies, to make 1000 simulations from the model reported in Section 3.2.2, ie., simulate new GIR values, including the estimated between study random variation on the parameters. Each time, the study is simulated using the exact same dose values and number of rats on HI and the insulin analogue as in the original data on this study. The following structure from the two studies were used in the simulations: study 040801 with six doses of HI (4, 6, 6, 7, 6, 6 rats) and six doses of the insulin analogue (5, 6, 6, 6, 8, 6 rats), study 060906 with three doses of HI (4, 5, 4 rats) and three doses of the insulin analogue (6, 5, 5 rats). The second step is to analyse each of the 1000 simulated studies (we now have 1000 simulated studies for each of the two chosen historical studies) using the three different modelling approaches from Section 3.2.

When performing the mixed-effects meta-analysis as well as the Bayesian analysis, we investigate how removing different numbers of control rats from the study affects the parameter of interest, $\log \rho$. Simulating the studies makes it easy to experiment with removing control rats, which is done by removing rats one at a time, such that the rats already removed do not enter the dataset again when decreasing the number of rats further. The standard approach is performed on each of the 1000 simulated datasets without removing any rats, and this is considered as our benchmark results on $\log \rho$ and its standard error for that study. Lastly, we can then examine these simulated results to investigate if including historical information, either by the mixed-effects meta-analysis approach or the Bayesian approach, might result in reducing the number of control rats without compromising on precision compared to the standard method.

## 4.2 Results from the mixed-effects meta-analysis approach

Using the mixed-effects meta-analysis approach, it can be discussed whether to include historical information only for HI or also historical information on the insulin analogues. As described in Section 3.2.2, it is clearly the inclusion of historical HI data that will be of highest importance, but the results actually become more precise and do not change dramatically when including the insulin analogues as well, which is why we choose this approach for our simulation experiment. The results from the simulation experiment for $\log \rho$ are seen in Table 2. Note that the parameter estimates based on the original data can be found in Table 1.

The seven estimates from Table 2 are based on the 1000 simulations, where *stan* indicates that the standard analysis is performed on each of the simulated datasets. The rest of the subscripts indicate how many control rats are removed from the simulated data when re-fitting by the mixed-effects meta-analysis approach. *none* and *all* indicate keeping and removing all the control rats, respectively, in *rem1* we remove one control rat per dose value, in *rem2* we remove two and so on. The two studies have different numbers of dose values, so for instance in study 060906 we have three different dose values for HI, so removing one rat per dose value means a reduction of three rats in this study.

For both studies the estimates remain stable whether performing the standard analysis or removing different numbers of rats in the mixed-effects meta-analysis approach, as seen in Table 2. However, the standard deviation on the estimated parameters does change as seen both in Table 2 and in Fig 4.

The figure indicates that the medians for both studies remain stable using the different ways of re-fitting the simulated data. The box plots for both studies show that removing all control rats results in a too large variation on the parameter compared to the red boxplot, which marks the standard analysis. For study 060906, the boxplots indicate that the relative log-potencies when removing around 1–2 control rats per dose resemble the tendency in the box plot for the corresponding standard analysis. Similarly, we get the same picture when removing around 3 control rats for study 040801.

Even if removing the same number of rats per dose value, the number of control rats as well as dose values differ between the two studies, which leads to a difference in percent in the resulting reduction. In Fig 5, it is clarified exactly how much the reductions in rats per dose value correspond to in percent of the total number of control rats in the respective study.

Here, we have shown the standard error from performing the standard analysis with a red dotted line. The points are simply the standard errors from Table 2 for a different number of

**Table 2. Results from the mixed-effects meta-analysis based on 1000 simulations.** Estimates and standard errors for the relative log-potency of the insulin analogue compared to HI for study 060906 (insulin analogue 32) and 040801 (insulin analogue 7) are shown for different ways of re-fitting the simulated data: none and all indicate keeping and removing all the control rats, respectively, rem1 indicate removing one control rat per dose value, rem2 indicate removing two and so on.

| | Study 060906 | | Study 040801 | |
|---|---|---|---|---|
| | **Estimate** | **Sd** | **Estimate** | **Sd** |
| $\log \hat{\rho}_{stan}$ | -0.412 | 0.132 | -0.620 | 0.105 |
| $\log \hat{\rho}_{none}$ | -0.410 | 0.126 | -0.621 | 0.097 |
| $\log \hat{\rho}_{rem1}$ | -0.410 | 0.130 | -0.626 | 0.099 |
| $\log \hat{\rho}_{rem2}$ | -0.412 | 0.136 | -0.622 | 0.102 |
| $\log \hat{\rho}_{rem3}$ | -0.413 | 0.144 | -0.621 | 0.106 |
| $\log \hat{\rho}_{rem4}$ | -0.412 | 0.158 | -0.619 | 0.114 |
| $\log \hat{\rho}_{remAll}$ | -0.409 | 0.170 | -0.621 | 0.152 |

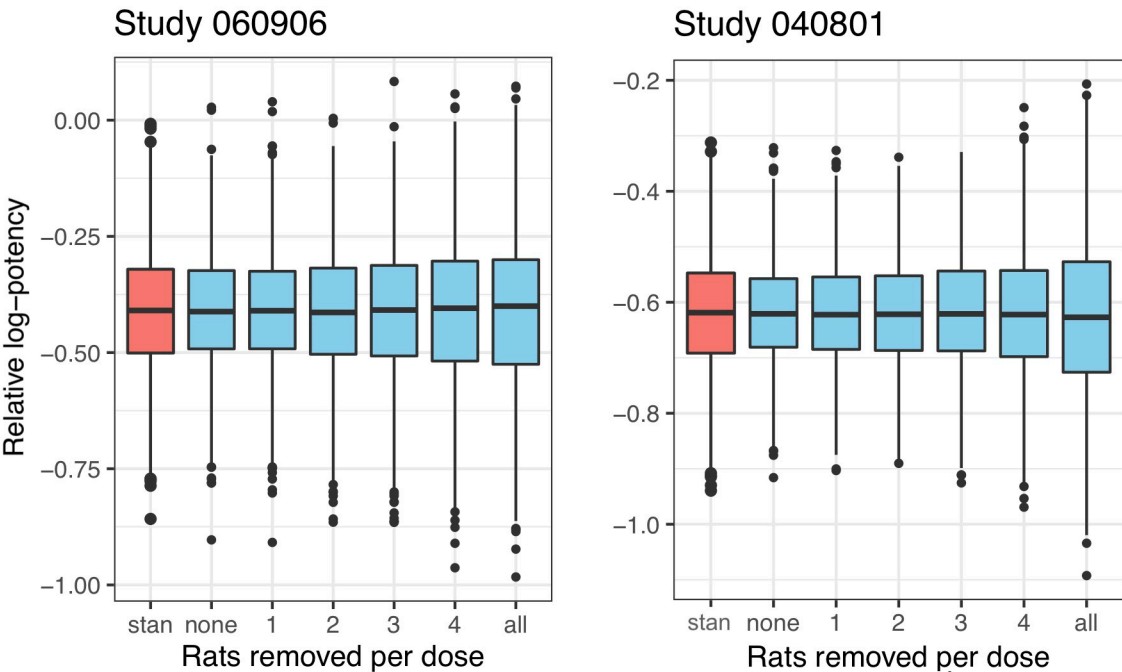

**Fig 4. Box plots of the estimated relative log-potency based on 1000 simulations for the two studies.** The red box indicates a re-fit using the standard approach, and the remaining blue box plots are the re-fits using the mixed-effects meta-analysis approach, removing different numbers of control rats per dose value: none and all indicate keeping and removing all the control rats, respectively, rem1 indicates removing one control rat per dose value, rem2 removing two and so on.

control rats removed per dose value, and in the legend to the right, we have indicated the resulting reduction in percent of the total number of control rats.

Concerning study 060906, removing somewhere in between one and two rats per dose would lead to obtaining the same standard error as in the standard analysis, which we also concluded from Fig 4. So, to obtain lower or the same standard error, we are able to remove one rat per dose. This study has three dose levels with a total number of 13 control rats, thus we can remove $3/13 \approx 23\%$ of the control rats, which is also indicated in Fig 5 in the legend to the right. For study 040801, we are able to remove two control rats per dose, and removing three gives almost the same standard error as the standard approach. Removing two and three control rats results in a reduction of $2 \cdot 6/35 \approx 34.3\%$ and $3 \cdot 6/35 \approx 51.4\%$ respectively. Thus, by investigating an insulin analogue using this approach, we could potentially use half of the control rats on HI and still avoid compromising on precision compared to the standard approach.

### 4.3 Results from the Bayesian analysis

The results from the previous section show that including historical information seems beneficial if we want to be able to reduce the number of control rats on HI in future studies. The results from the Bayesian alternative described in Section 3.2.3 are shown in Table 3 for the two studies, removing a different number of control rats as described in Section 4.1. Concerning the estimates, these are almost identical to those from the mixed-effects meta-analysis reported in Table 2. When comparing the standard errors with Table 2, it is quite difficult to spot the difference, and therefore Figs 4 and 5 for this Bayesian alternative approach yield such similar figures that they are not shown here. Since we obtain such similar results for all three

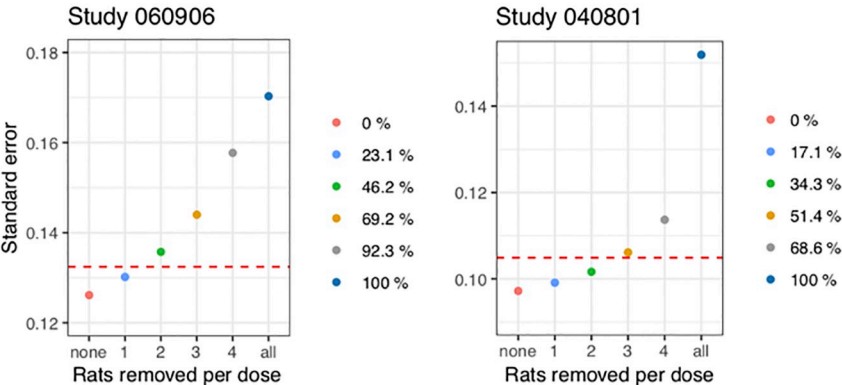

**Fig 5. Standard error for the relative log-potency based on 1000 simulations for the two studies.** The red dotted line is the standard error on the parameter from re-fitting the simulated data using the standard approach. The points indicate the re-fits using the mixed-effects meta-analysis approach, removing different numbers of control rats. On the x-axis, the number of control rats removed per dose is seen, and the legend shows the resulting reduction in percent of the total number of control rats.

**Table 3. Results from the Bayesian analysis based on 1000 simulations.** Estimates and standard errors for the relative log-potency of the insulin analogue compared to HI for study 060906 (insulin analogue 32) and 040801 (insulin analogue 7) are shown for different ways of re-fitting the simulated data: none and all indicate keeping and removing all the control rats, respectively, rem1 indicate removing one control rat per dose value, rem2 indicate removing two and so on.

|  | Study 060906 |  | Study 040801 |  |
| --- | --- | --- | --- | --- |
|  | **Estimate** | **Sd** | **Estimate** | **Sd** |
| $\log \hat{\rho}_{stan}$ | -0.412 | 0.132 | -0.620 | 0.105 |
| $\log \hat{\rho}_{none}$ | -0.410 | 0.126 | -0.619 | 0.097 |
| $\log \hat{\rho}_{rem1}$ | -0.411 | 0.131 | -0.620 | 0.099 |
| $\log \hat{\rho}_{rem2}$ | -0.412 | 0.137 | -0.618 | 0.102 |
| $\log \hat{\rho}_{rem3}$ | -0.413 | 0.144 | -0.617 | 0.107 |
| $\log \hat{\rho}_{rem4}$ | -0.414 | 0.160 | -0.619 | 0.115 |
| $\log \hat{\rho}_{remAll}$ | -0.411 | 0.171 | -0.620 | 0.153 |

studies compared to the mixed-effects meta-analysis approach, we can conclude that this Bayesian approach might be used as an alternative way of incorporating the historical information.

## 5 Discussion

Using the standard method as a benchmark, in Section 4 we investigated the possibility of reducing the number of control rats including historical information and using either a mixed-effects meta-analysis approach or a Bayesian approach, where the latter is based on a construction of a model for the historical data. Due to a high degree of homogeneity across the historical studies, both approaches suggest that including the previous studies is beneficial in regard to reducing the number of control rats when conducting a new study. However, both approaches rely heavily on the mixed-effects model to be used, and therefore the importance of building a suitable model and investigating possible heterogeneity across studies must be underlined and assessed if necessary. In our case, the studies are similar giving a high homogeneity, but this might not be true in other experiments. As discussed in [4], some established

Bayesian methods to discount the effect of a possible heterogeneity in the parameter estimates are power prior, commensurate prior, and meta-analytic predictive prior. The Bayesian approach used in this article essentially adapts what is called the meta-analytic-predictive approach in [4], namely exploiting the series of historical studies at hand in order to construct an informative prior. The "robust" extension of this method, that serves to account for data where the difference between historical and current data exceeds the heterogeneity among the historical trials, or a power prior, might have been beneficial to apply in our case. However, these methods require additional parameters, which would have been challenging to pre-define or estimate. The power prior requires a parameter for weighing the historical studies compared to the current study. For a hierarchical data structure with a time series of historical studies, it is not straightforward how to choose or even estimate this parameter from data. This would require different weights depending on the distance in time between the studies, giving the most recent studies the largest weight. The development of parameter estimates over time in our series of studies has been monitored, see Fig 3, which would also be needed in other experiments, in order to be aware of a possible larger heterogeneity as time passes. If information from some historical studies is not "accurate" in the sense that outlier studies are observed, it should be handled carefully, in order to justify the quality and effects for making valid statistical inference based on this historical information. This could be done by removing or down-weighing some studies (after careful consideration), for example by implementing a variant of a power-prior, using different pre-defined weights.

If the mixed-effects model is based on data from too few studies, the between study variation cannot be accurately estimated [10], and by following the standard approach from Section 3.2.1, we avoid this challenge, as the concern about "the absolute faith one has to place in the multivariable model" is avoided [11]. The standard method might therefore be somehow advantageous, but building a model that includes relevant historical information could possibly shorten the length of a new study, as we induce the possibility of including fewer control rats to get the same level of precision as in the standard approach of analysing the data, see results from Section 4. Here, it is important to be aware that including historical controls should be done carefully, ensuring that there is no reason to suspect systematic differences between these [10]. This was shown to be the case for the historical control rats in our dataset, by thoroughly building the mixed-effects model which was presented in Section 3.2.2.

In the simulation experiment, when using the mixed-effects meta-analysis approach to re-fit the data, we are sure that the model is updated according to the available data, which one can argue gives the most reliable results. Conversely, in the Bayesian analysis this new data is not automatically taken into account in the model, and there is thus a need to decide how often the mixed-effects model underlying the Bayesian analysis should be updated. But how big is the need really to update the model? In [1], they conclude that only a modest number of studies is sufficient since not much gain is obtained by continuously increasing the number of historical studies. Therefore, if future studies continue behaving similarly to previous studies, the effect of adding a new study when building the model can be difficult to spot. So, there is no clear answer to which approach gives most valid results, but in general, as they also state in [1], our results in Section 4 show that precision for estimating treatment effect can be improved by using historical studies, which leads us to the final conclusion.

## 6 Conclusion

In this paper, the possibility of reducing the number of control rats by including historical information from previous studies was investigated. A nonlinear logistic curve for the dose-response relationship was presented and used as the foundation for the different modelling

approaches. A carefully built mixed-effects model including all available historical data was used in a simulation experiment with a focus on the results of the relative log-potency for the insulin analogues under investigation compared to HI for two randomly chosen historical studies. In this experiment, the standard approach was used as the benchmark results, ie. including data only from the study under investigation to keep in mind how the studies were normally analysed.

In the simulation experiment, when using the mixed-effects model presented, the estimate of the relative log-potency did not change for the two insulin analogues examined in depth, compared to the benchmark results from the standard analysis. However, most interestingly, the uncertainty concerning these parameters decreased, as indicated by the standard error dropping. So, because HI exhibited similar behaviour across the studies, we got a more precise estimate using this method. This simulation experiment suggested overall that by including the historical information in the form of the mixed-effects model proposed, we were able to remove between 23% and 51% of the control rats in the two studies to get the same level of precision on the relative log-potency as in the standard analysis. How to incorporate the historical information in the form of the mixed-effects model was discussed. Both a mixed-effects meta-analysis approach as well as a Bayesian approach was suggested, and the above conclusions were found to be similar for the two methods. Thus, it can be concluded that the inclusion of historical information is beneficial in regard to using fewer control rats.

## Supporting information

**S1 Appendix. R-code for the different modelling approaches.**
(PDF)

**S1 Dataset. Dataset analysed in this paper consisting of a time series of 59 glucose clamp studies.** The dataset is provided by Novo Nordisk A/S.
(XLSX)

## Author Contributions

**Conceptualization:** Søren Andersen.

**Data curation:** Christian Lehn Brand.

**Formal analysis:** Emilie Prang Nielsen.

**Investigation:** Emilie Prang Nielsen.

**Methodology:** Emilie Prang Nielsen, Søren Andersen, Susanne Ditlevsen.

**Project administration:** Emilie Prang Nielsen, Christian Lehn Brand.

**Supervision:** Søren Andersen, Susanne Ditlevsen.

**Visualization:** Emilie Prang Nielsen.

**Writing – original draft:** Emilie Prang Nielsen.

**Writing – review & editing:** Emilie Prang Nielsen, Søren Andersen, Christian Lehn Brand, Susanne Ditlevsen.

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
