## [Decision Letter · Decision Letter 0]

21 Apr 2022

PONE-D-21-28972Applying Historical Data in a Nonlinear Mixed-Effects Model Can Reduce the Number of Control Rats Required for Calculation of the Relative Potency of Insulin AnaloguesPLOS ONE

Dear Dr. Nielsen,

Thank you for submitting your manuscript to PLOS ONE. After careful consideration, we feel that it has merit but does not fully meet PLOS ONE’s publication criteria as it currently stands. Therefore, we invite you to submit a revised version of the manuscript that addresses the points raised during the review process.

The manuscript has been evaluated by two reviewers, and their comments are available below.

The reviewers have raised a number of concerns that need attention. They request additional information on methodological aspects of the study, the study discussion, the data availability and more.

Could you please revise the manuscript to carefully address the concerns raised?

We look forward to receiving your revised manuscript.

Kind regards,

Sebastian Shepherd

Associate Editor

PLOS ONE

Journal Requirements:

3. Thank you for stating the following in the Competing Interests section: "I have read the journal's policy and the authors of this manuscript have the following competing interests: Søren Andersen and Christian Lehn Brand are employed by Novo Nordisk and hold shares in the company."

Reviewers' comments:

Reviewer's Responses to Questions

**Comments to the Author**

1. Is the manuscript technically sound, and do the data support the conclusions?

Reviewer #1: Yes

Reviewer #2: Yes

2. Has the statistical analysis been performed appropriately and rigorously? 

Reviewer #1: Yes

Reviewer #2: Yes

3. Have the authors made all data underlying the findings in their manuscript fully available?

Reviewer #1: Yes

Reviewer #2: No

4. Is the manuscript presented in an intelligible fashion and written in standard English?

Reviewer #1: Yes

Reviewer #2: Yes

5. Review Comments to the Author

Reviewer #1: The use of external controls from similar studies has been a popular topic mainly among statisticians in clinical trials. These statistical methods show the promise of designing more efficient trials, particularly, in settings of small sample size and slow enrolment. Animal studies are limited by sample size due to cost, logistics, or ethical reasons. I commend the authors of this paper for demonstrating the utility of using historical controls in this interesting and important application. The authors have carefully chosen similar historical studies for information borrowing. A nonlinear mixed-effects model was used to assess the dose-response relationship of insulin analogues as compared to human insulin. The relative log-potency was estimated using a meta-analytic approach and a Bayesian approach. Both approaches yielded comparable results and demonstrated that it would be possible to reduce the number of control animals in a study while borrowing this information from historical studies conducted under similar protocols. Adequate guidance and caution has been provided as to when and how this analysis should be conducted. Although the approach and application are sound, I do have some edits and suggestions to further improve the presentation:

1. It would be helpful to the general readers if the authors followed the journal’s citation guideline:

“References are listed at the end of the manuscript and numbered in the order that they appear in the text. In the text, cite the reference number in square brackets (e.g., “We used the techniques developed by our colleagues [19] to analyze the data”). PLOS uses the numbered citation (citation-sequence) method and first six authors, et al.”

2. In accordance with scientific writing, “meta approach” should be spelled out as “meta-analysis approach” throughout the manuscript.

3. Page 1, paragraph 3: Please, clarify the sentence “Sufficient similarity between historical and current control data is also a breaking point for.9”

4. Section 2: Please, include a sentence explaining why there are 69 analogues from 59 studies.

5. Page 5, paragraph 1: The parameters are estimated by maximum likelihood or Markov chain Monte Carlo methods ….

6. Page 9: “markov chain monte carlo” should be “Markov chain Monte Carlo”.

7. Results from the Bayesian method should be reported as in Table 1.

8. In the discussion, the authors have commented that it was possible to achieve higher precision by combining studies “due to a high degree of homogeneity across the historical studies”. This is the ideal case and may not be true in other experiments. The degree of heterogeneity need to be assessed and appropriate methods need to be used to discount the effect of this heterogeneity in the parameter estimates. This is an additional advantage of Bayesian methods. Some established methods are power prior, commensurate prior, and meta-analytic predictive prior as discussed in reference [5]. It would be worthwhile to include this information in the discussions.

Reviewer #2: In this paper, the authors investigated the possibility of reducing the number of control rats by using available historical information from the previous studies. Three different procedures within the nonlinear logistic curve framework are considered. Numerical results from simulation studies and real-data application support that the inclusion of historical information is valuable and beneficial. The paper is well-written and I enjoy reading of this paper. My specific comments are as follows.

(1) In this paper, the authors considered both the meta procedure and Bayesian procedure for this study. I wonder if the author consider power prior Bayesian analysis for the current study, in which the power prior is adopted to take the amount of information from historical data into consideration. If so, I strong suggest the authors consider this procedure.

(2) Given that the data is not available and may not publicly available, I wonder how the results will be replicated even though the authors have made the Code available.

(3) What will happen for the proposed procedures if the historical information is not accurate. In addition, I wonder how the authors deal with the heterogeneity issue of the different studies.

(4) In addition, how do we justify the quality and effects of historical information for making statistical inference quantitatively?

6. PLOS authors have the option to publish the peer review history of their article (what does this mean?). If published, this will include your full peer review and any attached files.

Reviewer #1: No

Reviewer #2: **Yes: **Min Wang

---

## [Author Response · Author response to Decision Letter 0]

20 May 2022

Dear Reviewers and Editors, 

Please find answers to review comments and journal requirements in the "Response to Reviewers" document.

Sincerely,

Emilie

---

## [Editor Report · Decision Letter 1]

3 Jun 2022

Applying historical data in a nonlinear mixed-effects model can reduce the number of control rats required for calculation of the relative potency of insulin analogues

PONE-D-21-28972R1

Dear Dr. Nielsen,

Thank you for submitting your manuscript titled “Applying historical data in a nonlinear mixed-effects model can reduce the number of control rats required for calculation of the relative potency of insulin analogues” to PLOS ONE. 

In this paper, the authors investigated the possibility of reducing the number of control animals by incorporating historical control information from the previous studies that used similar protocols. A nonlinear mixed-effects model was used to assess the dose-response relationship of insulin analogues as compared to human insulin. The relative log-potency was estimated using a meta-analytic approach and a Bayesian approach. The results from simulation studies and real-data application support that the inclusion of historical information is valuable and beneficial. Adequate guidance and caution has been provided as to when and how this analysis should be conducted.

The manuscript was reviewed by myself along with another independent reviewer. Both reviewers approved the manuscript for its technical soundness, rigor of statistical methods, and the writing and presentation style. The reviewers had few comments on the use of standard statistical terminology, data availability, and a discussion on alternative approaches when there is heterogeneity in historical studies. The authors have adequately addressed all comments and met all journal requirements. Therefore, the manuscript is now formally accepted for publication in PLOS ONE.

Kind regards,

Nusrat Harun

Guest Editor

PLOS ONE
---

## [Editor Report · Acceptance letter]

8 Jun 2022

PONE-D-21-28972R1 

Applying historical data in a nonlinear mixed-effects model can reduce the number of control rats required for calculation of the relative potency of insulin analogues 

Dear Dr. Nielsen:

I'm pleased to inform you that your manuscript has been deemed suitable for publication in PLOS ONE. Congratulations! Your manuscript is now with our production department. 

Kind regards, 

on behalf of

Dr. Nusrat Harun 

Guest Editor

PLOS ONE